

# A new dataset of rain cells generated from observations of the Tropical Rainfall Measuring Mission (TRMM) precipitation radar and visible and infrared scanner and microwave imager

**Zhenhao Wu[1], Yunfei Fu[1], Peng Zhang[2], Songyan Gu[2], and Lin Chen[2]**

[1]School of Earth and Space Sciences, University of Science and Technology of China, Hefei, 230026, China

[2]Key Laboratory of Radiometric Calibration and Validation for Environmental Satellites, National Satellite Meteorological Center, China Meteorological Administration, Beijing, 100081, China

**Correspondence:** Yunfei Fu (fyf@ustc.edu.cn)

**Abstract.** Rain cells are the most common units in the natural precipitation system. Enhancing the understanding of these rain cell characteristics can significantly improve the cognition of the precipitation system. Previous studies have mostly analyzed rain cells from a single radar data. In this study, we merged the precipitation parameters measured by the Tropical Rainfall Measuring Mission (TRMM) precipitation radar (PR) with the muti-channel cloud-top radiance measured by the visible and infrared scanner (VIRS) and the muti-channel brightness temperature measured by the TRMM microwave imager (TMI). The rain cells were identified within the PR orbit, and the swath truncation effect was eliminated. We used two methods for rain cell identification: the minimum bounding rectangle (MBR) method and the best fit ellipse (BFE) method, and compared the differences between these two methods in describing the rain cell characteristics. The results indicate that both methods can better reflect the geometric characteristics of rain cells. Compared with the MBR method, the BFE method can obtain a smaller rain cell area, and the filling ratio is better. However, the MBR method can simplify the data storage volume. Consequently, we employed the MBR method to analyze the precipitation structure of two typical rain cell precipitation cases. The results show that the new rain cell dataset can be used for the analysis of rain cell precipitation parameters and visible/infrared and microwave signals, which provides valuable data for comprehensive studies on the rain cell structural characteristics and furthers the understanding of precipitation mechanisms. The data which were used in this paper are freely available at https://doi.org/10.5281/zenodo.8352587 (Wu et al., 2023).

## 1 Introduction



Precipitation is an important part of the global energy and water cycle (Houze, 1997; Oki and Kanae,
2006; Lau and Wu, 2010). Rain cells are considered as the most basic precipitation unit, but they have
been defined differently in literatures. Austin and Houze (1972) studied the precipitation patterns in New
England and found that there are basically composed of subsynoptic scale precipitation regions, each
with rather clearly definable characteristics and behavior. Based on radar observations and detailed rain
gauge records, the precipitation patterns can be divided into synoptic areas, large mesoscale areas, small
mesoscale areas and cells. The rain cell with an area of about 10 $km^2$ radar echo is regarded as a single
cumulus convective unit in their study. Goldhirsh and Musiani (1986) defined rain cell as areas with rain
rates greater than or equal to the threshold by setting a corresponding rain rate threshold. Meanwhile,
many papers studied the relationship between precipitation rate threshold and rain cell size based on
ground-based radar data (Konrad, 1978; Sauvageot et al., 1999; Feral et al., 2000; Begum and Otung,
2009). Capsoni et al. (1987) defined rain cell as a connected region with a precipitation rate greater than
5 $mm\ h^{-1}$ based on S-band radar observation near Milan in 1980, and obtained the relationship between
the spatial number densities of rain cells and the cumulative distribution of precipitation rate. Awaka
(1989) modified the precipitation rate threshold to 0.4 $mm\ h^{-1}$.
In order to obtain the shape parameters, Feral et al. (2000) employed the elliptic fitting method to
investigate the geometric characteristics and directional distribution of rain cells. The statistical results
revealed that the major axis length was twice longer than the minor axis length for the majority of the
rain cells, and the direction distribution was uniform. When studying the horizontal dimensions of the
precipitation systems, Nesbitt et al. (2006) used the best-fit ellipse method to obtain the major axis and
minor axis. However, due to the limitation of the swath width, some precipitation systems were truncated.
The results indicated that there were obvious differences of storm morphology characteristics over land
and ocean, which may lead to significant differences in regional precipitation simulation. Liu and Zipser
(2013) investigated the different horizontal structures of convective precipitation systems in the tropics
and subtropics by using 14 years of Tropical Rainfall Measuring Mission (TRMM) Precipitation Radar
(PR) observations combined with the best ellipse fitting method. They found that line shaped convective
systems occurred more frequently over ocean, and showed higher frequency in the subtropics.
Fu et al. (2020) used the minimum bounding rectangle (MBR) method to identify rain cells and studied
the geometric and physical parameter characteristics of rain cells over tropical land and ocean areas with
15-yr measurements of the TRMM PR. The study showed that the average rain rate of rain cells is more



frequently related to the increase of area, and the increasing rate over land is greater than that over ocean.
Chen et al. (2021) used the same dataset in the above study to analyze the multidimensional
morphological characteristics of precipitation areas over the Tibetan Plateau in summer, and found that
there is a close relationship between the morphological characteristics of precipitation areas and the
intensity of precipitation.
The investigation of the three-dimensional structure of rain cells is helpful to understand the
thermodynamic structure and microphysical processes within precipitation systems (Houze, 1981; Zipser
and Lutz, 1994; Yuter and Houze, 1995; Liu and Fu, 2001). The PR onboard the TRMM provides an
excellent opportunity to study the 3D structure of precipitation (Kummerow et al., 1998, 2000; Nesbitt
et al., 1999; Schumacher and Houze, 2003; Li and Fu, 2005), and the visible and infrared scanner (VIRS)
on the satellite can provide spectral signals and cloud parameters information on the top of precipitation
clouds (Liu and Fu, 2010; Fu, 2014), the TRMM microwave imager (TMI) can provide information of
different phase particles inside precipitation clouds (Viltard et al., 2000; He et al., 2006; Fu et al., 2021).
Fu et al. (2007a) analyzed the characteristics of precipitating and non-precipitating clouds in typhoon
Ranan occurred in August 2004 by matching and merging data measured by TRMM PR, TMI and VIRS.
The result indicated that large particles are mostly in the precipitation cloud, and the effective radius
distribution of non-precipitating cloud particles is relatively wide. Fu et al (2007b) utilized the results of
multi-instrument measurements on TRMM to analyze several typical precipitation systems over the East
Asia, and reveal the relationship of precipitation structure, lightning activities, precipitation cloud top
and rainfall rate near surface. Liu et al (2008) established a precipitation feature database based on 9
years of TRMM observation data, matching the VIRS and TMI observation results to PR pixels resolution.
Subsequently, Liu et al (2009) used this database to study the contribution of warm rain systems to
precipitation over the tropics and obtained the seasonal and spatial distribution of warm rain systems.
This paper is merging the PR, VIRS, and TMI measurements at PR pixel resolution, and combining
the rain cell identification method to establish a new precipitation parameter and visible/infrared and
microwave signal dataset. Section 2 describes the data and merging methods. Section 3 introduces the
definition method of rain cells and identification method. Section 4 defines the geometric and physical
parameters of rain cells. In Section 5, the statistical results of rain cell parameters are given and the
structures of two typical rain cells are analyzed. Access to the datasets is introduced in Sect. 6, and
conclusions are presented in Sect. 7.

## 2 Data

### 2.1 Tropical Rainfall Measuring Mission

The TRMM was jointly developed by the US National Aeronautics and Space Administration (NASA) and the Japan Aerospace Exploration Agency (JAXA) and launched on November 27, 1997. The TRMM is a non-solar synchronous polar-orbiting satellite with an orbital inclination of 35° and observes a location between 38°S and 38°N (Simpson et al., 1996; Kummerow et al., 1998,2000). The satellite carries five instruments: PR, VIRS, TMI, the Lighting Imaging Sensor (LIS), the Cloud and Earth Radiant Energy Sensor (CERES). This study mainly involves the measurements of TRMM PR, VIRS, and TMI.

### 2.2 PR and 2A25 dataset

The PR was the first spaceborne precipitation radar onboard the TRMM. It is a single-frequency microwave radar with a frequency of 13.8 GHz (Kummerow et al., 1998; Kozu et al., 2001). PR scans in the cross-track direction with a scanning inclination of 17°. There are 49 pixels on each scanning line. The horizontal resolution is about 4.3 km at nadir (5.0 km after the orbital boost), and the scanning width is 215 km (245 km after the orbital boost). It can detect the three-dimensional structure of precipitation from mean sea level to 20 km (a total of 80 layers) with a vertical resolution of 0.25 km.

The 2A25 data is the second-level data product of the TRMM PR, which is generated by inverting the echo signals detected by the PR. This dataset mainly includes scanning time, geographic information, three-dimensional rain rate, rain type and so on (Awaka et al., 1997). The detection sensitivity of the PR is about 17 dBZ, corresponding to the rain rate of about 0.4 $mm\,h^{-1}$ (Schumacher and Houze, 2003). Therefore, when the rain rate of the pixels is lower than 0.4 $mm\,h^{-1}$, the default value is set and will not be involved in the calculation.

### 2.3 VIRS and 1B01 dataset

The VIRS scans in the cross-track direction with a scanning angle of 45°. There are 261 pixels on each scanning line. The scanning width is 720 km (833 km after the orbital boost), and the horizontal resolution is 2.2 km at nadir (2.4 km after the orbital boost). It has five channels from visible to the far infrared band: CH1 (0.63 μm), CH2 (1.6 μm), CH3 (3.7 μm), CH4 (10.8 μm) and CH5 (12.0 μm).

The 1B01 is a first-level data product of VIRS, which includes the reflectivity (RF1, RF2) and the infrared radiation brightness temperature ($TB_{3.7}$, $TB_{10.8}$, $TB_{12.0}$) after the correction and calibration of



the VIRS detection results.

### 2.4 TMI and 1B11 dataset

The TMI is a nine-channel passive microwave radiometer with five frequencies spanning from 10 to 85
GHz. The microwave signal frequencies are 10.65, 19.35, 21.3, 37.0, and 85.5 GHz, except for 21.3 GHz,
which is a single vertical polarization channel. The other four frequencies are horizontal (H) and vertical
(V) polarization dual channels. The scanning width is 760 km (878 km after the orbital boost). The
horizontal resolution of each frequency channel (effective field of view of beam, Kummerow et al., 1998)
varies from 63 km × 37 km at 10.65 GHz to 7 km × 5 km at 85.5 GHz. The 1B11 data contains the
calibrated TMI-detected microwave brightness temperature at multiple channels.

### 2.5 2A25, 1B01 and 1B11 merged data

To comprehensively analyze the parameters of precipitation, cloud top spectral signal and particle phase
in the cloud precipitation system, the 2A25, 1B01 and 1B11 data products (derived from the TRMM PR,
VIRS and TMI, respectively) were matched and merged. Due to the difference in data detection methods,
the spatial resolution is different for the three instruments, but the time lag between detections of the
same target is less than 1 min. It provides the possibility of data merging because of the quasi-
synchronous detection. In this paper, we default to the consistency of time during data merging, only
considering spatial merging. Based on the pixel resolution of PR detection, VIRS and TMI are matched
to corresponding pixels according to latitude and longitude (Liu et al., 2008; Fu et al., 2011; Sun and Fu,

142 2021).

For the PR and VIRS, in order to obtain the spectral signals (reflectivity and infrared radiation
brightness temperature) near the precipitation cloud top, the horizontal resolution of 1B01 pixel is
decreased to make it consistent with 2A25. Generally, there are usually about 7 VIRS pixels near 1 PR
pixel. The specific method is to introduce the Gaussian function. The spectral signals are calculated by
weighted averaging the VIRS near the PR pixel. It was found that the spectral signals changed weakly
after merging, the mean change was less than 0.7%, and the mean square deviation was less than 2.5%
(Fu et al., 2011). Sun and Fu (2021) considered the merging process between the PR and VIRS pixels
was no dramatic variations on the original data, and the conclusions were consistent with Fu et al. (2011).
For TMI, the measured values in different bands have different spatial resolutions. In order to obtain
the microwave radiation brightness temperature in PR pixel resolution, it is necessary to increase the

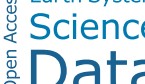

horizontal resolution of 1B11 pixels to make it consistent with the resolution of 2A25 pixels. By using
the nearest neighbor method, each PR pixel is assigned a TMI pixel closest to it (Liu et al., 2008).

## 3 Algorithm of rain cell identification

Previous studies have pointed out that the measurements of PR are inevitably affected by the swath
truncation effect due to the limitation of the PR detection orbit. Nesbitt et al. (2006) analyzed 3-year
TRMM PR data and concluded that nearly 42% of total rainfall is affected by the swath truncation. Liu
and Zipser (2013) showed that due to the influence of swath truncation, the parameters described rain
cell would be distorted and the captured precipitation systems would be incomplete. Fu et al. (2020)
defined rain cell as neighboring rain pixels with at least 0.4 $mm\ h^{-1}$ near-surface rain rate or 17 dBZ
reflectivity by using the PR data. The rain cells affected by the swath truncation were removed in the
study, and the rain cells completely detected by PR swath were retained.
This paper also adopts the same definition and elimination method of rain cells proposed by Fu et al.
(2020). Firstly, using the eight-connected region method identifies the continuous region of rain pixels.
For example, for any rain pixel A, iteratively search its neighboring pixels (a total of 8 pixels). If one or
more of them are also rain pixels (marked as B, C, ⋯), A, B, C and ⋯ are identified as the same rain
cell. Then, the search process is repeated for B, C, ⋯ rain pixels until all adjacent pixels of any rain
pixel in the rain cell are non-precipitation pixels. Upon completion of the identification process for a rain
cell, the identified rain cell is numbered. Secondly, the rain pixel swath truncation identification and the
elimination of micro-rain cells (less than 4 pixels) are executed. The identified rain cells are
systematically traversed. When a rain cell edge is identified as located at the edge of the track, it is
defined as a truncated rain cell. If the number of pixels in a rain cell is less than 4, it is defined as a micro
rain cell. The above rain cells will be removed (Chen, 2019).
To describe the shape of the rain cell, the minimum bounding rectangle (MBR) rain cell identification
method and the best fit ellipse (BFE) rain cell identification method are used to identify the rain cell. The
MBR rain cell identification method is to find the rectangle with the smallest area covering the target
object (rain cell) by rotating the external rectangle (Fu et al., 2020). The BEF rain cell identification
method fits the most suitable elliptical shape wrapped rain cell according to the polygonal connection
frame of the outer boundary of the rain cell (Nesbitt et al., 2006).

## 4 Definitions of parameters of rain cell



We define some geometric and physical parameters to describe the identified rain cells (Fu et al., 2020).
The specific geometric parameters are listed in Table 1, where the first six parameters are called the
horizontal geometric parameters and the rest are the vertical geometric parameters. The differences in
geometric parameters of the MBR method and the BFE method are studied. Where $\alpha$ represents the
horizontal shape of the rain cell, a small (large) $\alpha$ indicates that the two-dimensional shape of the rain
cell is more like a strip (square) system, and has more (less) correlation with a frontal system. The
variable $\beta$ expresses the ratio of the rain cell area ($S_{rain}$) to the area of the identification frame ($S$), and
characterizes the internal organization within the rain cell. Large (small) $\beta$ suggests that those rain
pixels are more (less) compactly organized, and are more (less) likely to be associated with strong
convective systems. The variables $\gamma_{max}$ and $\gamma_{av}$ represent the three-dimensional spatial shape of the
rain cell. Small $\gamma_{max}$ ($\gamma_{av}$) indicates a "squatty" appearance of rain cell, in contrast to a "lanky"
appearance for large $\gamma_{max}$ ($\gamma_{av}$).

**Table 1. Definitions of geometric parameters of rain cell.**

| Symbol | Geometric meaning of rectangle | Geometric meaning of ellipse |
|---|---|---|
| $L$ (km) | $r_L$ (km) | $e_L$ (km) |
| | Length of the MBR method | Length of the major axis of the BFE method |
| $W$ (km) | $r_W$ (km) | $e_W$ (km) |
| | Width of the MBR method | Width of the minor axis of the BFE method |
| $\alpha$ | $r_\alpha$ | $e_\alpha$ |
| | Horizontal shape of rain cell, $r_\alpha = r_W/r_L$ | Horizontal shape of rain cell, $e_\alpha = e_W/e_L$ |
| $S_{rain}$ (km²) | Area of the rain cell, sum of all areas of rain pixels | |
| $S$ (km²) | $r_S$ (km²) | $e_S$ (km²) |
| | Area of the MBR method, $r_S = r_L * r_W$ | Area of the BFE method, $e_S = \frac{\pi}{4} * e_L * e_W$ |
| $\beta$ | $r_\beta$ | $e_\beta$ |
| | Filling ratio of the rain cell, $r_\beta = S_{rain}/r_S$ | Filling ratio of the rain cell, $e_\beta = S_{rain}/e_S$ |
| $H_{max}$ (km) | Maximum echo top height among rain pixels in the rain cell | |
| $H_{av}$ (km) | Mean echo top height averaged among rain pixels in the rain cell | |
| $\gamma_{max}$ | Maximum spatial morphology, $\gamma_{max} = H_{max}/L$ | |
| $\gamma_{av}$ | Mean spatial morphology, $\gamma_{av} = 2H_{av}/(L+W)$ | |
| $H_{avc}$ (km) | Mean echo top height of convective precipitation in the rain cell | |
| $H_{avs}$ (km) | Mean echo top height of stratiform precipitation in the rain cell | |


Since the rain cell dataset contains the measurements of multiple instruments, we define some physical
parameters (Table 2) based on rain type, rain rate profiles, near surface rain rates, visible reflectivity and
infrared brightness temperature. Those parameters are significant to represent the intensity, the





inhomogeneity, and the evolution stage of rain cells.

**Table 2. Definitions of physical parameters of rain cell.**

| Symbol | Physical meaning |
|---|---|
| $RR_{ave}$ (mm h$^{-1}$) | Mean rain rate obtained by averaging all rain rates within the rain cell |
| $RR_{max}$ (mm h$^{-1}$) | Maximum rain rate among rain pixels of the rain cell |
| $RR_{avc}$ (mm h$^{-1}$) | Mean convective rain rate averaged by all convective rain rates within rain cell |
| $RR_{avs}$ (mm h$^{-1}$) | Mean stratiform rain rate averaged by all stratiform rain rate within rain cell |
| $RR_{maxc}$ (mm h$^{-1}$) | Maximum rain rate among convective pixels of the rain cell |
| $RR_{maxs}$ (mm h$^{-1}$) | Maximum rain rate among stratiform pixels of the rain cell |
| CAF (%) | Fraction of convective area to total precipitation area within the rain cell |
| SAF (%) | Fraction of stratiform area to total precipitation area within the rain cell |
| CPC (%) | Convective precipitation contribution to total precipitation within the rain cell |
| SPC (%) | Stratiform precipitation contribution to total precipitation within the rain cell |
| $dBZ_{max}$ (dBZ) | Maximum radar reflectivity factor among rain pixels of the rain cell |
| $H\_dBZ_{max}$ (km) | Height of the maximum radar reflectivity factor among rain pixels of the rain cell |
| $RF_{ave}$ | Mean reflectivity obtained by averaging all reflectivities of channel within the rain cell |
| $RF_{avc}$ | Mean convective reflectivity averaged by all convective reflectivities of channel within the rain cell |
| $RF_{avs}$ | Mean stratiform reflectivity averaged by all stratiform reflectivities of channel within the rain cell |
| $TB_{ave}$ (K) | Mean brightness temperature obtained by averaging all brightness temperatures of channel within the rain cell |
| $TB_{avc}$ (K) | Mean brightness temperature averaged by all convective brightness temperatures of channel within the rain cell |
| $TB_{avs}$ (K) | Mean brightness temperature averaged by all stratiform brightness temperatures of channel within the rain cell |


# 5 Results

## 5.1 Statistics of rain cell parameters

In this study, we select two methods for identifying rain cells. It is important to note that the choice of
identification method may influence the results of parameters. Taking the two rain cells shown in Figure
1 as example, they locate on the southern slope of the Tibetan Plateau (orbit 08691 on 2 June, 1999) and
the eastern part of the Tibetan Plateau (orbit 31787 on 13 June, 2003), respectively. Table 3 shows the
calculated horizontal geometric parameters of the rain cells displayed in Fig. 1.



The first case shown in Fig. 1a is calculated by the MBR method with $r_L$ = 290.86 km, $r_W$ = 140.29
km, $S_{rain}$ = 10223.5 km², $r_S$ = 40803.36 km², and $r_\beta$ = 0.25, which shows a strip shape ($r_\alpha$ = 0.48).
For the second case (Fig. 1b), $r_L$, $r_W$, $S_{rain}$, $r_S$, and $r_\alpha$ are 169.57 km, 76.18 km, 4496.4 km²,
12917.48 km², and 0.45, respectively. In addition, the filling ratio in the second case (0.35) is slightly
larger than that in the first case (0.25), resulting in a more compact rain cell system overall.

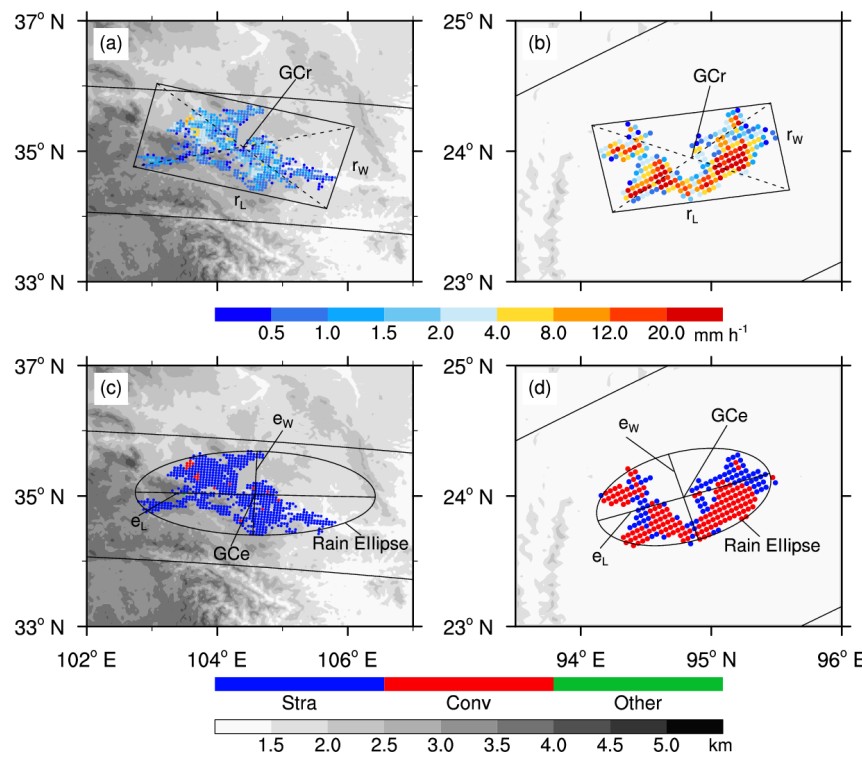


**Figure 1. Near surface rain rate of two rain cells that occurred on 2 June 1999 (a) and on 13 June 2003 (b)**
**measured by PR. In (c, d), rain pixels with blue, red, and green represent stratiform, convective, and other**
**precipitation, respectively. The identification boxes are captured by the MBR method (a, b) and the BFE**
**method (c, d) for two rain cells.**

The first case (Fig. 1c) is calculated by the BFE method with $e_L$ = 347.63 km, $e_W$ = 139.94 km, $e_S$
= 38207.27 km², $e_\alpha$ = 0.4, and $e_\beta$ = 0.27. For the second case (Fig. 1d), $e_L$, $e_W$, $e_S$, $e_\alpha$, and $e_\beta$ are
170.66 km, 76.76 km, 10289.43 km², 0.45, and 0.44, respectively. Based on the horizontal geometric
parameters of rain cells captured by the MBR and BFE method, it is evident that the area of the
rectangular box is larger than that of the ellipse box. This difference in area results in a smaller filling



ratio for the MBR method as compared to the BFE method. There is no significant difference between
the two methods in terms of other geometric parameters. In practical application, compared with the BFE
method, the MBR method can greatly simplify the amount of data storage. Moreover, the MBR method
can envelop all the pixels of the rain system, which works better than the BFE method in bow
precipitation and small precipitation system (Chen, 2019).

**Table 3. The horizontal geometric parameters of the first/second case calculated by MBR and BFE.**

| Rectangle | | Ellipse | |
|---|---|---|---|
| $r_L(km)$ | 290.86/169.57 | $e_L(km)$ | 347.63/170.66 |
| $r_W(km)$ | 140.29/76.18 | $e_W(km)$ | 139.94/76.76 |
| $r_\alpha$ | 0.48/0.45 | $e_\alpha$ | 0.4/0.45 |
| $S_{rain}(km^2)$ | 10223.50/4496.40 | $S_{rain}(km^2)$ | 10223.50/4496.40 |
| $r_S(km^2)$ | 40803.36/12917.48 | $e_S(km^2)$ | 38207.27/10289.43 |
| $r_\beta$ | 0.25/0.35 | $e_\beta$ | 0.27/0.44 |


The results of the parameter calculations shown in Table 4 are not affected by the rain cell identification
method. For the first case, the $H_{max}$ = 8.75 km, $H_{av}$ = 5.59 km, $\gamma_{max}$ = 0.03, and $\gamma_{av}$ = 0.026 show
that the vertical scale of the rain cell is at least one order of magnitude smaller than the horizontal scale.
The rain cell has $RR_{ave}$ = 1.27 and $RR_{max}$ = 8.08 mm h$^{-1}$, respectively. The parameters of CAF =
2.6 %, SAF = 97.3 %, CPC = 11% and SPC = 89% indicate that the rain cell is primarily composed of
stratiform precipitation. The maximum radar reflectivity factor (dBZ$_{max}$) is 36.38 dBZ, and the relative
height (H_dBZ$_{max}$) is 3.25 km. According to the merged data of VIRS and TMI instruments, the $RF1_{ave}$,
$TB_{10.8\_ave}$, $TB_{19GHz\_H\_ave}$, $TB_{85GHz\_H\_ave}$ are 0.72, 253.65 K, 260.61 K, and 254.44 K, respectively.
These parameters indicate that the stratiform rain cell has less precipitation and weak development.

**Table 4. The vertical geometric and physical parameters of the first/second case.**

| Vertical parameter | | Physical parameter | | | |
|---|---|---|---|---|---|
| $H_{max}(km)$ | 8.75/17.75 | $RR_{ave}$ (mm h$^{-1}$) | 1.27/11.64 | $RF1_{ave}$ | 0.72/0.69 |
| $H_{av}$ (km) | 5.59/9.47 | $RR_{max}$ (mm h$^{-1}$) | 8.08/113.14 | $RF1_{avc}$ | 0.73/0.66 |
| $\gamma_{max}$ | 0.03/0.11 | $RR_{avc}$ (mm h$^{-1}$) | 5.52/17.35 | $RF1_{avs}$ | 0.72/0.73 |
| $\gamma_{av}$ | 0.03/0.08 | $RR_{avs}$ (mm h$^{-1}$) | 1.16/2.31 | $TB_{10.8\_ave}$ (K) | 253.65/222.96 |
| $H_{avc}$ (km) | 5.76/10.39 | $RR_{maxc}$ (mm h$^{-1}$) | 8.08/113.14 | $TB_{10.8\_avc}$ (K) | 252.42/221.51 |
| $H_{avs}$ (km) | 5.58/7.96 | $RR_{maxs}$ (mm h$^{-1}$) | 4.45/11.87 | $TB_{10.8\_avs}$ (K) | 253.68/225.31 |
| | | CAF (%) | 0.03/0.62 | $TB_{19GHz\_H\_ave}$ (K) | 260.61/276.75 |
| | | SAF (%) | 0.97/0.38 | $TB_{19GHz\_H\_avc}$ (K) | 258.38/275.79 |
| | | CPC (%) | 0.11/0.92 | $TB_{19GHz\_H\_avs}$ (K) | 260.66/278.32 |



| | | | |
|---|---|---|---|
| SPC (%) | 0.89/0.08 | $TB_{85GHz\_H\_ave}$ (K) | 254.44/219.11 |
| $dBZ_{max}$ (dBZ) | 36.38/57.81 | $TB_{85GHz\_H\_avc}$ (K) | 252.84/212.72 |
| $H\_dBZ_{max}$ (km) | 3.25/2.75 | $TB_{85GHz\_H\_avs}$ (K) | 254.49/229.53 |


For the second case, the $H_{max}$ and $H_{av}$ are 17.75 km and 9.47 km. The parameters $\gamma_{max}$ = 0.11 and
$\gamma_{av}$ = 0.08 are slightly larger than those in the first case, indicating that the second rain cell is slightly
taller than the first one. The parameters $RR_{ave}$ and $RR_{max}$ are 11.64 and 113.14 mm h$^{-1}$ at this rain
cell, respectively, which are much larger than those in the first case. In addition, CPC (SPC) in the second
case has a value of 92% (8%), larger (smaller) than that in the first case. This indicates that the rain cell
in the first case has a greater proportion of stratiform precipitation in the total precipitation (89%), while
the second rain cell has a higher proportion of convective precipitation (92%). The parameters $dBZ_{max}$
and $H\_dBZ_{max}$ are 57.81 dBZ and 2.75 km, respectively. Compared with the first case, this case has
more precipitation and strong convective regions locate in lower layers. Furthermore, $RF1_{ave}$ has a
value of 0.69, slightly larger than that in the first case. Except for $TB_{19GHz\_H\_ave}$ = 276.75 K, the
$TB_{10.8\_ave}$ = 222.96 K and $TB_{85GHz\_H\_ave}$ = 219.11 K are smaller than the first case. This indicates that
the brightness temperature near the cloud top of the rain cell is relatively low, while the convection
develops vigorously and there is a higher concentration of ice particles within the cloud.

## 5.2 Structure of rain cell


In order to further understand the rain cells, the structural distribution of rain cells is given in this section.
We also compare the distribution of relevant parameters between the original datasets and the merged
datasets. Figure 2 presents the signals detected by the visible, and thermal infrared channels of VIRS
onboard TRMM. The solid black line represents the TRMM PR scan track, while the solid red line is the
rectangular box of the rain cell. The reflectivity at 0.63 μm is primarily related to cloud optical thickness
(COT), and the higher the COT value, the greater the reflectivity. The equivalent brightness temperature
of a blackbody at 10.8 μm indicates the height of cloud top, and the higher the cloud top, the lower the
brightness temperature of the infrared channel (Luo et al., 2020).

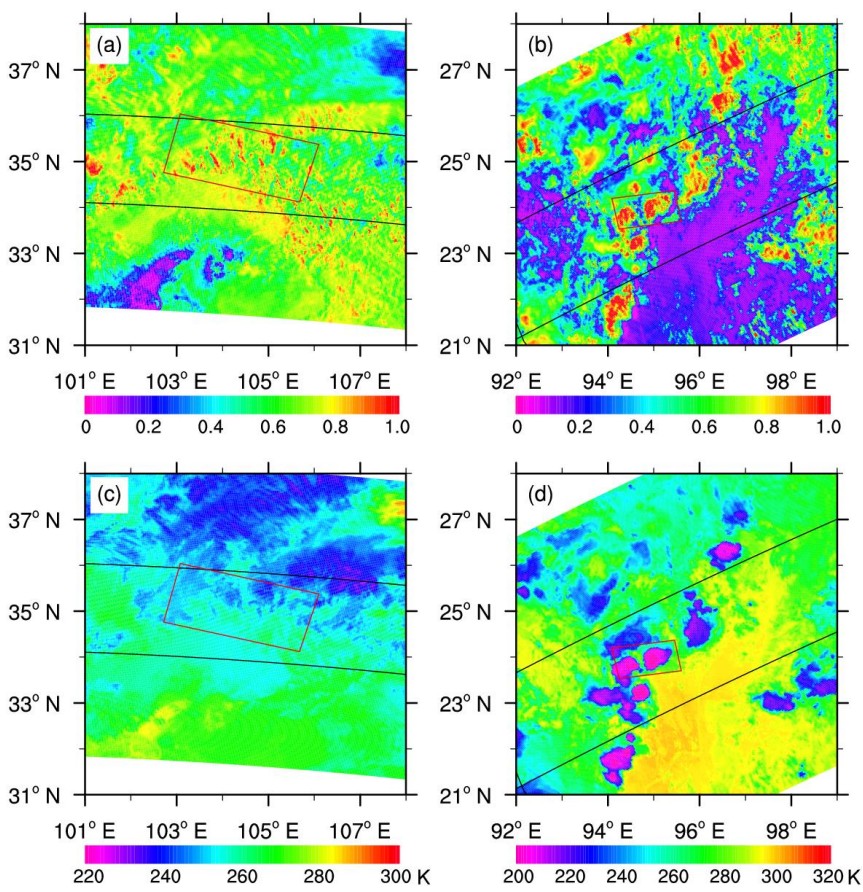

**Figure 2. The distribution of (a, b) RF1 and (c, d) TB$_{10.8}$ for two precipitation cases. The solid black line is the TRMM PR scanning track and the solid red line is the rectangular box of rain cell.**

In Fig. 2a, the reflectance of the visible channel in the rectangular box generally exceeds 0.6, with certain areas exhibiting values above 0.8. This suggests that the COT of certain areas of the stratiform rain cell is large. In addition, the brightness temperature of the 10.8 μm is mainly between 240-260 K (Fig. 2c), with low cloud development. There are two central regions with RF1 higher than 0.8 and corresponding TB$_{10.8}$ below 220 K (Figs. 2b, d). This suggests that there are many large-sized and deep ice cloud droplets dominating in the central region of the convective rain cell, with high cloud development. This analysis indicates that the signals obtained by TRMM/VIRS has strong detection ability for the cloud-top structure, and can have a good understanding of the development of the rain cell.

According to the theory of microwave remote sensing, the frequency of 10-23 GHz is sensitive to liquid water particles in the cloud, and the frequency of 36-89 GHz is sensitive to liquid and solid



particles (ice or snow) in the cloud (Petty, 1994a, 1994b; Fu et al., 2021). Figure 3 shows the brightness
temperature distribution of the 19 GHz and 85 GHz horizontal polarization channels observed by the
TMI. The outline of rain cells cannot be clearly observed for the two cases in Figs. 3a, b, because the
emission signal of precipitation particles in the cloud is drowned by the surface emission signal, which
is why passive microwave low-frequency signals cannot invert the land surface precipitation (Fu et al.,
2021). Figure 3c shows $TB_{85GHz\_H}$, which ranges between 240 and 260 K. The outline of the rain cell
can be roughly observed. The $TB_{85GHz\_H}$ of two convective central zones in rain cell are around 200-
220 K (Fig. 3d), indicating that ice-phase particles are more abundant in the convective precipitation
clouds.

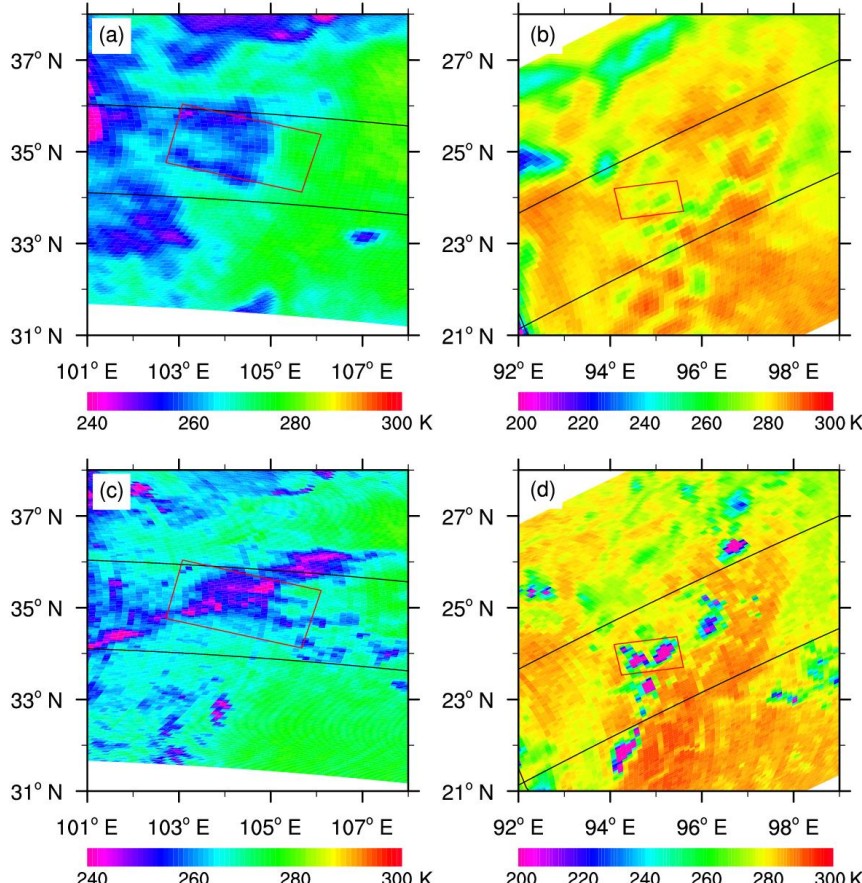

**Figure 3. The same as Fig. 2, but for the (a, b) $TB_{19GHz\_H}$ and (c, d) $TB_{85GHz\_H}$.**

Figure 4 shows the distribution of parameters in the rain cells merged dataset. The stratiform rain cell



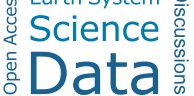

shown in the first case (Fig. 4a) has a uniform distribution of precipitation, with the maximum rain rate
exceeding 4 mm h$^{-1}$. However, the convective rain cell in the second case (Fig. 4b) presents a non-
uniform distribution, with large local variations, and the maximum rain rate can reach more than 20
mm h$^{-1}$. RF1 in the first case ranges between 0.6 and 0.8 (Fig. 4c). The value in the second case can
exceed 0.8 in the area of heavy precipitation, and the corresponding COT is larger (Fig. 4d). $TB_{10.8}$
varies from 240 to 260 K in the first case, and the cloud top height of rain cell is uniformly distributed
(Fig. 4e). While the $TB_{10.8}$ in the center of heavy precipitation varies between 200-220 K, which
suggests that the cloud top of the center region develops at a high altitude, whereas the altitude of the
cloud system around it is relatively low (Fig. 4f). $TB_{19GHz\_H}$ for two cases cannot observe valid
information due to the surface emissivity. The $TB_{85GHz\_H}$ varies uniformly from 240-260 K, which is
close to the distribution of the $TB_{10.8}$ (Fig. 4i). The region in the center of the strong precipitation in
Fig. 4j is below 200 K, which indicates that the cloud interior contains more ice-phase particles. In a
word, the merging process has no dramatic variations on the distribution of original data.

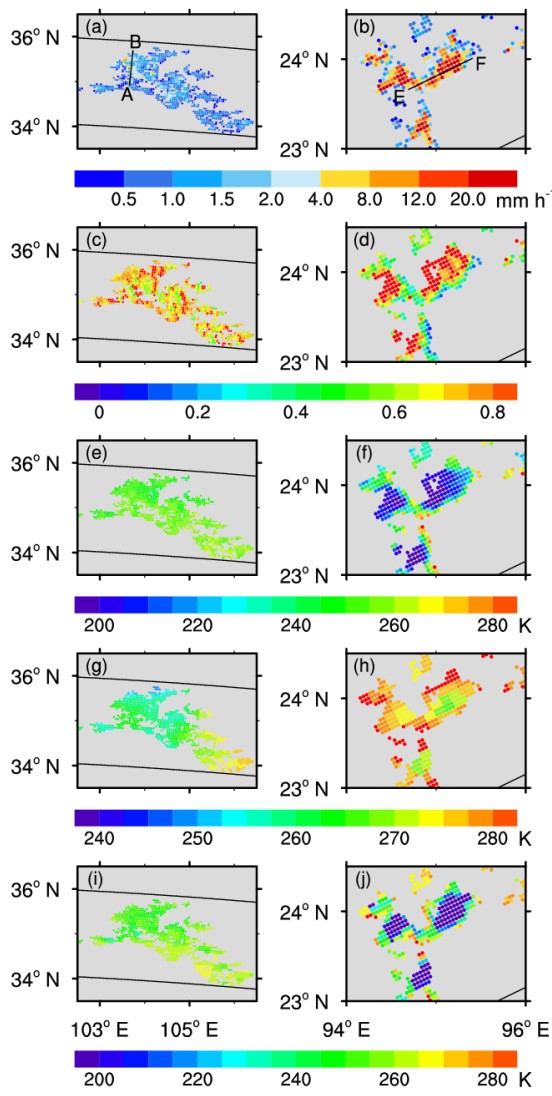

**Figure 4. The distributions of (a, b) the near-surface rain rate, (c, d) RF1, (e, f) $TB_{10.8}$, (g, h) $TB_{19GHz\_H}$, and (i, j) $TB_{85GHz\_H}$ for two precipitation cases based on the merged datasets.**

The distribution of radar reflectivity factor along the AB (as shown in the Fig. 4a) profile in Fig. 5a reveals that stratiform precipitation does not extend vertically beyond 8 km, with some areas exhibiting stratification. A shallow bright band is observed at a height of 4 km. The radar reflectivity factor of the precipitation is generally below 32 dBZ, indicating weak precipitation activity. Figure 5c shows that the reflectivity of the 0.63 μm channel varies mainly between 0.5 and 0.8. The 1.6 μm channel varies between

0.1 and 0.3, with these low signal levels further suggesting the presence of numerous ice-phase regions.
The change in the brightness temperature profile shows that the top of the stratiform precipitation cloud
has a higher temperature and correspondingly lower cloud top height (Fig. 5e). The microwave brightness
temperature values vary uniformly between 250 and 270 K, which may be strongly influenced by the
surface emissivity and thus may not serve as an accurate indicator (Fig. 5i).

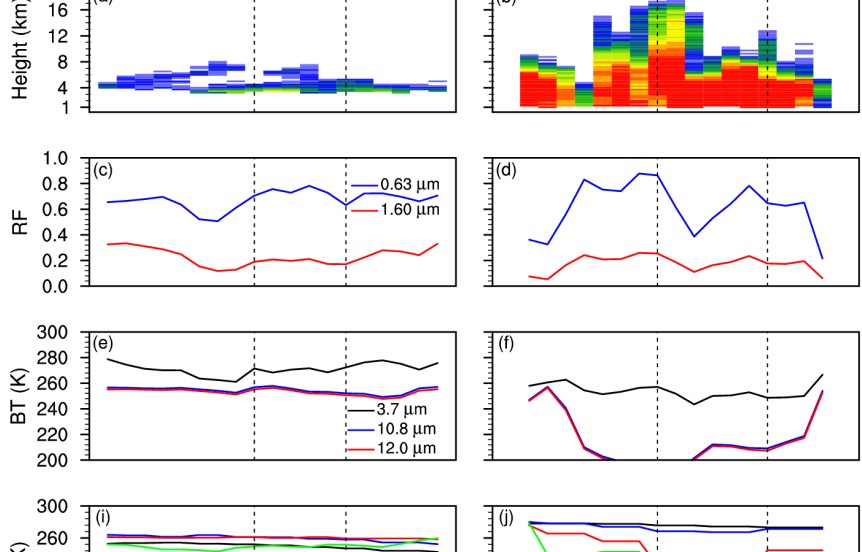


**Figure 5. The vertical cross sections of (a, b) the radar reflectivity factor, (c, d) reflectivity, (e, f) infrared**
**brightness temperatures, (i, j) microwave brightness temperature along the A-B and E-F line as shown in Fig**
**4.**

Figure 5b shows that the radar reflectivity factor presents an uneven distribution in both vertical and
horizontal directions due to the irregular movement of precipitation particles in the precipitation cloud
with the airflow. The echo top height can reach up to 17 km, which also means that strong convection
occurs here. In addition, the radar reflectivity factor of near-surface exceeds 38 dBZ, indicating heavy
precipitation near the surface. RF1 varies from 0.3 to 0.9, which means that the COT is unevenly
distributed. RF2 does not vary strongly, suggesting that the phase distribution in the cloud is uniform and
mostly ice-phase particles (Fig. 5d). The brightness temperature of far-infrared in the region of strong





convective development is lower than the other locations (Fig. 5f). Meanwhile, the height of the cloud
top is higher in the corresponding region. $TB_{10GHz\_H}$ and $TB_{19GHz\_H}$ vary uniformly and does not
respond well to precipitation. The low value area of the $TB_{37GHz\_H}$ and $TB_{85GHz\_H}$ tend to correspond
well to areas of strong precipitation, as shown by the location of the dotted dashes in Fig. 5j. There are
often filled with a high concentration of ice-phase particles in the areas of strong precipitation, leading
to strong scattering signal.

## 6 Data availability


The rain cell precipitation parameters and visible/infrared and microwave signal datasets used in this
paper are accessible at https://doi.org/10.5281/zenodo.8352587.

## 7 Discussion and conclusions


We establish a new rain cell precipitation parameter and visible infrared and microwave signal dataset
combining with the multi-instrument observation data on TRMM. PR provides the three-dimensional
precipitation structure of the rain cell, VIRS provides the spectral signal of the top of the rain cell, and
TMI provides the phase information of the particles inside the rain cell. The purpose of this dataset is to
promote the three-dimensional study of rain cell precipitation system, and reveal the spatial and temporal
variations of the scale morphology and intensity of the system.
First, we compare the difference between the MBR rain cell identification method and the BFE rain
cell identification method in describing rain cell shape. The two methods are mainly different in the
geometric parameters such as $L$, $W$, $\alpha$, $S$, and $\beta$. According to the parameter calculation results, both
methods can reflect the geometric characteristics of rain cells effectively. Compared with the MBR
method, the BFE method can obtain the fitting result of the smallest rain cell area and its $\beta$ value is
correspondingly larger than that of the MBR method. However, we adopt the MBR method to identify
rain cells in the new dataset because it can simplify the data storage volume.
Second, we calculate other parameters of two types of rain cell, including vertical geometric
parameters (average cloud top height, horizontal scale, spatial morphology, and so on) and physical
parameters (average rain rate, maximum reflectivity factor and its corresponding height, average visible
reflectivity, average brightness temperature, and so on). The statistical results show that the defined





parameters can effectively reflect the three-dimensional morphology of rain cells and the development
state of precipitation system, which can provide reference for exploration of the relationship between
rain cell morphology parameters and precipitation.
Third, we use the new dataset to analyze the distribution of precipitation parameters, cloud top
radiation signal, and brightness temperatures and the profile characteristics of two rain cells. The study
shows that merged dataset can significantly display the features of the original dataset. For the stratiform
rain cell, the rain rate is relatively small, with high cloud top brightness temperature, low cloud top height,
and weak vertical movement. For the convective rain cell, the rain rate is large and the distribution is
uneven with low cloud top brightness temperature, high cloud top height, and strong vertical movement.
The difference of parameter distribution between two types of rain cells is obvious, which lays a
foundation for the subsequent study on the characteristics of precipitation.
To further explore the characteristics of rain cell, the cloud parameters based on the signal retrieval
from the TRMM VIRS data and the latent heat of precipitation inside the rain cell will be added to the
dataset. This work is currently in progress but will not be covered in this study due to the limited length
of the paper. The dataset will support the research on the development mechanism of precipitation system
and promote the progress of precipitation model simulation in the future. With the continuous
development of satellite technology, the dataset will add longer time scale data and more effective
parameters.

**Author contribution.** ZW and YF prepared the data in the standardized format. ZW uploaded the data
in the data repository and prepared the manuscript with contribution from YF. All the authors discussed
the concepts and edited the manuscript.

**Competing interests.** The authors declare that they have no conflict of interest.

**Acknowledgements.** We would like to acknowledge the National Aeronautics and Space
Administration (NASA) for providing TRMM PR, VIRS and TMI datasets.

**Financial support.** This research has been supported by the National Natural Science Foundation of
China (grant nos. 42230612 and 42275140).

**Review statement.**





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
