# Peer review of "A new dataset of rain cells based on observations of"

_Earth System Science Data, 2023_

## Referee Comment (RC2)

A new dataset of rain cells generated from observations of the Tropical Rainfall Measuring Mission (TRMM) precipitation radar and visible and infrared scanner and microwave imager

Wu et al.

**Summary**

This paper presents a different method for defining "rain cells" using TRMM data, and publishes the new dataset for public use for meteorological studies. The method fuses TMI, PR, and VIRS data and uses a minimum bounding rectangle method to construct its features, as opposed to fitting ellipses, which has been the traditional method in the literature. Referring to other methods in the literature, there is already an extensive body of work constructing precipitation features with TRMM data, combining all three datasets. The authors reference this body of work but fail to describe how their dataset is an improvement beyond the state of the art. For this reason, I recommend rejection. I would recommend revision and resubmitting a paper in the same journal but selected as a method paper instead of a data paper, detailing how the MBR method is advantageous and presents a feature dataset beyond that which is already constructed by the Liu/Zipser groups and readily available.

**Major Comments**

Title: I would suggest removing the multiple "and"s and using commas

Literature: The background section reads as thoroughly as a textbook, going back to the 1970's for a definition of a "rain cell" – the references need to be updated to more relevant sources and some discussion of the motivation behind a consistent definition of a "rain cell" is needed, here. Different scales and boundaries are useful for different reasons.

This paper discusses, disjointedly, the Precipitation Feature Databases constructed by the Liu/Zipser groups at the University of Utah and Texas A&M Universities at Corpus Christi. The PF database has evolved into a massive undertaking with cells of every size and strength, and defined by PR radar (RPFs), and passive-microwave polarization corrected temperature (PCTFs). Some focusing solely on precipitation, convection, MCSs, tropical cyclones, etc. They are freely available for download from https://pps.gsfc.nasa.gov/ or http://atmos.tamucc.edu/trmm/.

While this manuscript references the dataset, it does not elucidate how the work advances the state of the art of constructing precipitation feature (or rain cell) databases, beyond trying a new boundary-definition method.

There are many other boundary definition methods currently used to define precipitating areas from satellite data, such as convex hull or K-means clustering. Many of the

Precipitation Features (PF) are defined using contiguous pixels, and do not require a bounding ellipse to be drawn.

It is not made clear in this manuscript how the authors' method advances this technique of defining features.

Minor Comments:

13 "Previous studies have mostly analyzed rain cells from a single radar data" This statement is false, there is a wealth of studies of rain cell (feature) database analysis from satellites.

17 Swath truncation: is this in reference to the ends of orbit files creating boundary artifacts? Or the edge of the swath on the sides cutting off features prematurely? It is not made clear in the rest of the manuscript.

28 Many journals do not allow citations in an abstract

33 suggest "the literature"

79 suggest "over East Asia"

83-84 How is what you have created different from previous datasets in the literature? Specifically, the PF database?

156 Please clarify if you are referring to the edge of the swath or the end of the orbit

---

## Author Comment (AC1)

**Responses to RC1**

We are grateful to the Editor and the Reviewers for reviewing our manuscript. The comments and suggestions are very helpful and valuable. According to the nice suggestions, we have made point-by-point response to the reviewers' comments.

**RC1:**

Main comments

1 Lines 21- 24. "Compared with the MBR method, the BFE method can obtain a smaller rain cell area, and the filling ratio is better. However, the MBR method can simplify the data storage volume. Consequently, we employed the MBR method to analyze the precipitation structure of two typical rain cell precipitation cases." How is difference in storage between two methods? Is difference of data storage between the two methods the only reason to choose MBR method?

Response:Thank you very much. This paragraph has been deleted. This question is answered in the conclusion of the manuscript: "It must be noted that the difference between MBR method and BFE method is only in the horizontal geometric parameters of the rain cell."[Line 313-314]

  The reason why two methods are used is also from scientific considerations, i.e. one more method is better than one. But your question is very good. Let's be clear about our starting point.

2 Fig. 1 shows that there are stratiform, convection and other precipitation. How was this classification made? Is it proved by the used datasets?

Response:Thanks. The identified precipitation types (convective precipitation, stratiform precipitation, other types of precipitation) were provided by the standard data of TRMM PR algorithm, and the three types of precipitation are the basic components of rain cell. The specific algorithm of precipitation type is complicated, which was

discussed in detail in papers listed below. Simply, stratiform precipitation was identified by bright-band, convective precipitation was identified by 40 dBZ threshold, and precipitation outside the two categories was other type of precipitation. Since the precipitation type identification method (V-method, Vertical profile method, H-method, Horizontal pattern method) had been proved correct by many studies (except for the Tibetan Plateau), we do not need to spend time to verify.

Since the defined rain cell was composed of different precipitation types, the characteristics of rain cell and its regional differences, changes in the precipitation process, or climate changes can be studied by analyzing the proportion and intensity of different precipitation types in the rain cell. Therefore, the rain cell data established in this study has important scientific significance for the cognition of the rain cell.

Please look the listed two references:

Awaka, J., Iguchi, T., and Okamoto, K.: TRMM PR standard algorithm 2A23 and its performance on bright band detection, J. Meteorol. Soc. Japan. Ser. II, 87A, 31–57, https://doi.org/10.2151/jmsj.87A.31, 2009.

Fu, Y. F., Liu, Y., Wu, Z. H., Zhang, P., Gu, S. Y., Chen, L., and Nan, S.: A new algorithm of rain type classification for GPM dual-frequency precipitation radar in summer Tibetan Plateau, Adv. Atmos. Sci., https://doi.org/10.1007/s00376-024-3384-7, 2024.

3 For the definitions of the rain cell, it is using a threshold of 17 dBZ. Is that mean that it will miss some weak cumulus cloud, e.g. cumulus without precipitation?

Response:Thank you very much. You asked very good questions! 17 dBZ is the threshold for TRMM PR to identify precipitation, which is determined by the performance of PR wavelength 2.2 cm (frequency 13.8 GHz). For small-scale particles, PR cannot scatter at its wavelength, so PR cannot identify such weak precipitation, so the rain cell we defined does not include such very weak precipitation.

In subsequent studies, we will use FY-3G PMR or GPM DPR data to establish rain

cells, because PMR and DPR have Ka-band radar, which can identify weak precipitation. TRMM PR, VIRS, and TRM were selected to build the rain cell data because they have 15 years of observations.

4 It was noticed that this paper is a data description paper. I agree that the increasing improvement of rain cell characteristics can promote our cognition of the precipitation system. The authors developed a new dataset for the rain cell. I think readers want to know what can we do with the new dataset. That says, in which aspect the new dataset can improve our knowledge of the cloud and precipitation system.

Response:That's a very good reminder. Thank you very much. In the conclusion of the revised manuscript, I have added a paragraph describing the use of the rain cell data: "The new rain cell data in this study can be used to study the peculiarity of rain cell geometric and physical parameters. Although a lot of achievements have been made in this aspect, systematic and in-depth analysis is still needed, such as the regional differences of these parameters and the characteristics of climate change. It can also be used to analyze the relationship between the physical and geometric parameters of rain cell, which also have regional differences. The effective radius of cloud particles, optical thickness, liquid water path and other parameters in rain cell can be obtained by combining retrieval algorithms of visible and near infrared reflectivity, which can analyze the characteristics of cloud physical parameters of rain cell. These parameters such as cloud water and ice water in column, cloud temperature and rain rate in rain cell can also be obtained by using microwave brightness temperature retrieval algorithms, and the relationship among these parameters can be analyzed." [Line 317-326]

5 The illustration of the new data via cases shown in Fig.1. How do the authors choose these cases and what is special characteristics of these cases? If we want to use the dataset to investigate the rain cell, what should we carefully concern?

Response:Thank you very much. These two rain cells were randomly selected, one had more weak precipitation (stratiform precipitation) and another more strong precipitation (convective precipitation). Usually, we have knowledge of the two kinds of precipitation systems. Compared with the geometric and physical parameters of the rain cell identified in this study, the results were also in line with our inherent cognition on the two kinds of precipitation system, such as the weak rain cell is not deep in the vertical direction with small rain rate, and the strong rain cell is deep with low infrared brightness temperature. The rain cell identification algorithm of this study will be provided to the National Satellite Meteorological Centre of CMA. It is believed that the rain cell data will be available to use in the near future, and detailed explanations will be given. There are no special precautions when using.

Minor comments

1 Line 20. I think word "well" is more proper than "better".

Response:Changes were made in the revised manuscript. Thanks.

2 The format of Table 1 should be refined. Text of column 2 occupies space of column 3.

Response:Thank you for your nice suggestion. Changes have been made in the revised manuscript.

All the above have been modified in the revised manuscript. Thank you again.

---

## Author Comment (AC2)

**Responses to RC2**

We are grateful to the Editor and the Reviewers for reviewing our manuscript. The comments and suggestions are very helpful and valuable. According to the nice suggestions, we have made point-by-point response to the reviewers' comments.

**RC2:**

This paper presents a different method for defining "rain cells" using TRMM data, and publishes the new dataset for public use for meteorological studies. The method fuses TMI, PR, and VIRS data and uses a minimum bounding rectangle method to construct its features, as opposed to fitting ellipses, which has been the traditional method in the literature. Referring to other methods in the literature, there is already an extensive body of work constructing precipitation features with TRMM data, combining all three datasets. The authors reference this body of work but fail to describe how their dataset is an improvement beyond the state of the art. For this reason, I recommend rejection. I would recommend revision and resubmitting a paper in the same journal but selected as a method paper instead of a data paper, detailing how the MBR method is advantageous and presents a feature dataset beyond that which is already constructed by the Liu/Zipser groups and readily available.

Response:Thank you very much for your frank comments and suggestions, but I regret your negative decision. I would like to reply you as follows, perhaps you will change your previous decision.

First of all, I want to tell you that the corresponding author has made significant rewrites to the original manuscript, because there were many unreasonable descriptions in the original expression. Secondly, we describe the starting point of this study, that is, the previous study results must be respected, such as the contributions made by Dr. Liu, Nesbitt, Zipser, et al. in this field. This is the scientific ethics that subsequent Scientific workers must follow, so the revised manuscript makes positive comments on their methods and results. In the revised manuscript, it is pointed out that the identified rain

cell is within the PR scanning range to ensure the integrity of the identified rain cell. We know that the rain cell data established by Dr. Liu/Zipser is very good and widely used. We often read the articles published by their group and are deeply inspired by them.

In our study, the proposed rain cell identification method and the corresponding data may be another choice for studies, after all, more datasets should be better than one for scientific research.

I hope you will be satisfied with my response to your comments. Since we are colleagues, I welcome you to communicate with us. Thanks again.

Major Comments

1 Title: I would suggest removing the multiple "and"s and using commas

Response: Thank you. The title has been revised.

2 Literature: The background section reads as thoroughly as a textbook, going back to the 1970's for a definition of a "rain cell" – the references need to be updated to more relevant sources and some discussion of the motivation behind a consistent definition of a "rain cell" is needed, here. Different scales and boundaries are useful for different reasons.

Response: Thanks. The revised manuscript was rewritten. The history of rain cell studies was mentioned to highlight the importance of Nesbitt, Liu and Zipser's work on the identification of rain cell "With the massive data observed by PR, VIRS and TMI, Nesbitt et al. (2006), Liu et al. (2007, 2008), Liu and Zipser (2013) made spick-and-span studies in the field of rain cell identification and its parameters with elliptic fitting method. Their rain cell data were also widely used on analyzing the temporal and spatial distribution characteristics of rain cell (Zhou et al., 2013; Yokoyama et al., 2014; Ni et al., 2015), such as that line shaped convective systems occurred more frequently over ocean, and showed higher frequency in the subtropics (Liu and Zipser, 2013)." [Line

55-60]

Because this study laid emphasis on the rain cell method (MBR and BFE) so the previous rain cell identification methods were briefly summarized, and those specific studies will be discussed later studies. Thanks again.

3 This paper discusses, disjointedly, the Precipitation Feature Databases constructed by the Liu/Zipser groups at the University of Utah and Texas A&M Universities at Corpus Christi. The PF database has evolved into a massive undertaking with cells of every size and strength, and defined by PR radar (RPFs), and passive-microwave polarization corrected temperature (PCTFs). Some focusing solely on precipitation, convection, MCSs, tropical cyclones, etc. They are freely available for download from https://pps.gsfc.nasa.gov/ or http://atmos.tamucc.edu/trmm/.

While this manuscript references the dataset, it does not elucidate how the work advances the state of the art of constructing precipitation feature (or rain cell) databases, beyond trying a new boundary-definition method.

There are many other boundary definition methods currently used to define precipitating areas from satellite data, such as convex hull or K-means clustering. Many of the Precipitation Features (PF) are defined using contiguous pixels, and do not require a bounding ellipse to be drawn.

It is not made clear in this manuscript how the authors' method advances this technique of defining features.

Response:Thank you for your frank questions. There is some ambiguity in the expression concerning your questions in our original manuscript. In the rewrite manuscript, we made drastic changes and additions, such as the use of data (see the last paragraph in the conclusion).

This study presents a slightly different approach to the rain cell from that used by Dr. Liu/Zipser's group. The mainly difference was rain cell identified within the scope of PR scan, and rain cell data of two cases was supplied.

As you said, it is a huge project to build a massive database that is easy for everyone

to use. The China Satellite Meteorological Center is already implementing this project, and we are also involved. As an independent study unit, programs need to be written by itself and methods needs to be established by itself. However, we can learn from the data provided by the international community and the methods already adopted, which is also a respect for the existing studies.

I hope my above answer will please you, because you have recognized that we have a spirit of independent study and respect for previous study results and scientists. Thank you very much again.

Minor Comments:

1 13 "Previous studies have mostly analyzed rain cells from a single radar data" This statement is false, there is a wealth of studies of rain cell (feature) database analysis from satellites.

Response:Thank you. It's missing from the rewrite.

2 17 Swath truncation: is this in reference to the ends of orbit files creating boundary artifacts? Or the edge of the swath on the sides cutting off features prematurely? It is not made clear in the rest of the manuscript.

Response:Thank you. Although this issue is not specifically discussed in the revised manuscript, we have been redescribed to avoid confusion. Please see line 130 to 133 "According to the working mode of PR, its swath consists of 49 pixels (from number 1 to 49), so if the identified rain cell has pixels at the edge of the PR swath (the first pixel and 49th pixel), the rain cell is not included. If the identified rain cell is at the beginning and end of the PR swath, the rain cell is also eliminated.", and line 137 to 139 "The slight differences of geometric parameters calculated by MBR method and BFE method show in the length and width of rain cell, while the physical parameters calculated by the both methods are the same.". Dr. Liu/Zipser's approach is not wrong either, we just have a slightly different focus. You know that very well.

3 28 Many journals do not allow citations in an abstract

Response:Thanks for reminding us, we will follow the requirements of ESSD.

4 33 suggest "the literature"

Response:Thank you. It's missing from the rewrite manuscript.

5 79 suggest "over East Asia"

Response:Thank you. It's missing from the rewrite manuscript.

6 83-84 How is what you have created different from previous datasets in the literature? Specifically, the PF database?

Response:Thank you. First of all, the data set created in this study has something in common with the PF database, both were built based on the TRMM multi-instrument observations. By defining different characteristic conditions, PF database forms various precipitation parameters, including microwave brightness temperature, infrared temperature and lightning information, which shows that the information of PF database is very comprehensive and worth to learn. The data set of this study took the rain cell within the swath of PR scanning as the object, defined its geometric and physical characteristics parameters. The identified rain cells were numbered and convenience search. The specific differences between our dataset and the PF database will be further studied.

7 156 Please clarify if you are referring to the edge of the swath or the end of the orbit

Response:Thanks for reminding me. In revised manuscript, the sentence had been changed into "According to the working mode of PR, its swath consists of 49 pixels (from number 1 to 49), so if the identified rain cell has pixels at the edge of the PR swath (the first pixel and 49th pixel), the rain cell is not included. If the identified rain cell is at the beginning and end of the PR swath, the rain cell is also eliminated." [Line 131-133]

All the above have been modified in the revised manuscript. Thank you again.

---

## Referee Report (RR1)

This work provides a dataset that includes features derived from two types of fittings applied to rain cells observed by TRMM PR, along with collocated VIRS and TMI brightness temperatures. This dataset has potential value for users who wish to extract horizontal and vertical geometric information of precipitation clusters from the extensive TRMM dataset. However, the publicly available data, as indicated by the URL provided by the authors, is currently limited in temporal coverage. As a result, users are unable to conduct long-term analyses of cloud clusters, which is likely one of their primary interests. Additionally, some expressions in the manuscript are potentially misleading to readers, and the description of the data processing methods is inadequate. For these reasons, there are critical issues that must be addressed before this manuscript can be considered for publication.

**Major Comments:**

1.

The data description paper should be published only after a sufficient amount of data is made available for users. Although the Abstract and Section 6 ("Data availability") refer to the data sources, only two days' worth of data are currently available. Since the extraction of rain cell features is primarily intended for statistical analyses, long-term data is essential. The currently provided data is insufficient for effectively utilizing the dataset. While it is understandable that the original TRMM precipitation and brightness temperature datasets are large in volume. However, at minimum, the data under the "Rain" node in the dataset should be made available for a longer period.

2.

The manuscript contains several instances where the description could mislead readers regarding the processing performed on the dataset. These should be corrected.

For example, in the Abstract, the following sentences appear (Lines 14–19):

"*In this study, based on the merged data of precipitation profile data, reflectivity and infrared data, and microwave brightness temperature data, …, rain cells were identified in the PR swath by two methods, the minimum bounding rectangle (MBR) method and the best fit ellipse (BFE) method. The geometric and physical parameters of rain cell were also defined.*"

This wording suggests that rain cell identification was performed based on merged data from three sensors. However, as described in Section 3, rain cell identification is determined solely based on the PR data, by checking whether four or more pixels are connected. The infrared

and microwave brightness temperature data are used only for deriving the physical parameters after the identification. Moreover, the MBR and BFE methods do not perform identification; they are fitting methods applied to already-identified rain cells.

Other instances with potentially misleading wording include:
- Lines 70–73 ("For the above...")
- Lines 134–135 ("To automatically...")
- Lines 301–304 ("By matching...")

Furthermore, many other parts of the manuscript incorrectly refer to MBR and BFE as identification methods. To avoid confusion, I recommend consistently referring to them as fitting methods.

3.

In Section 5.2, while the two case studies help contextualize the dataset, the findings primarily reflect well-established knowledge and do not seem to introduce new insights. Please clarify what new synergies are expected by incorporating VIRS and TMI data into the dataset alongside the fitting of rain cells using the MBR and BFE methods. Additionally, since Section 5.2 feels somewhat redundant, I suggest minimizing the description of well-known results and moving them to the Introduction.

**Minor comments:**

Sections 3 and 4

They currently describe the algorithm separately; however, it would be more organized to combine them into a single section, such as a unified "Methods" section.

Table 2

It should be clearly indicated which sensor each variable is derived from.

Table 3

Many values listed in Table 3 are not referenced in the main text. It would be better to remove unnecessary items. Regarding $\gamma_{max}$ and $\gamma_{avg}$, should these not differ between the MBR and BFE methods? Also, for parameters where the values are identical between MBR and BFE, combining them into a single column would improve readability.

Lines 193–203

The content described in this paragraph might be easier to understand if explained directly with reference to Figure 1. If the authors wish to present specific numbers, it would be sufficient to mention them within the text. Therefore, I would suggest that Table 4 may not be necessary.

---

## Author Response (AR2)

Dear Editor,

Thank you very much for your great help. We greatly appreciate the reviewers for taking their valuable time to review the manuscript. We also thank the two reviewers for their suggestions and comments, which have been of great help in improving the quality of our manuscript. After carefully reading the good suggestions from the editor and reviewers, we have made point-by-point response to the reviewers' comments. Please refer to the revised manuscript for details. Below are our responses to each comment and suggestion.

Yunfei Fu

RC2:

This work provides a dataset that includes features derived from two types of fittings applied to rain cells observed by TRMM PR, along with collocated VIRS and TMI brightness temperatures. This dataset has potential value for users who wish to extract horizontal and vertical geometric information of precipitation clusters from the extensive TRMM dataset. However, the publicly available data, as indicated by the URL provided by the authors, is currently limited in temporal coverage. As a result, users are unable to conduct long-term analyses of cloud clusters, which is likely one of their primary interests. Additionally, some expressions in the manuscript are potentially misleading to readers, and the description of the data processing methods is inadequate. For these reasons, there are critical issues that must be addressed before this manuscript can be considered for publication.

Response: Thank you very much for your nice suggestion! Regarding the limited data availability on the website, we have updated the dataset. The data volume is relatively large because it is under the detection resolution of the PR orbit. In the updated dataset, we have provided a comprehensive dataset of rain cells for the summers of 1999 and 2003 (from June to August). This dataset generally meets the needs of users and can be

used to analyze the characteristics of summer rain cells in the middle and low latitudes before and after the TRMM ascending orbit. The data are freely available at https://doi.org/10.5281/zenodo.15387988. Furthermore, we have refined the description of the data processing methods in the article to enable readers to better understand the methods.

**Major Comments:**

1.

The data description paper should be published only after a sufficient amount of data is made available for users. Although the Abstract and Section 6 ("Data availability") refer to the data sources, only two days' worth of data are currently available. Since the extraction of rain cell features is primarily intended for statistical analyses, long-term data is essential. The currently provided data is insufficient for effectively utilizing the dataset. While it is understandable that the original TRMM precipitation and brightness temperature datasets are large in volume. However, at minimum, the data under the "Rain" node in the dataset should be made available for a longer period.

Response: Thank you for what you said. Based on your suggestion, we have uploaded the rain cell datasets for the summers (June–August) of 1999 and 2003 to the updated dataset website. We hope this meets your requirements. The new dataset enables long-term statistical analysis of summer precipitation in mid-low latitudes regions. Researchers can use this dataset to conduct in-depth studies on the structural characteristics of rain cells and precipitation mechanisms. The data are freely available at https://doi.org/10.5281/zenodo.15387988.

2.

The manuscript contains several instances where the description could mislead readers regarding the processing performed on the dataset. These should be corrected. For example, in the Abstract, the following sentences appear (Lines 14–19):

"In this study, based on the merged data of precipitation profile data, reflectivity and

infrared data, and microwave brightness temperature data, …, rain cells were identified in the PR swath by two methods, the minimum bounding rectangle (MBR) method and the best fit ellipse (BFE) method. The geometric and physical parameters of rain cell were also defined."

This wording suggests that rain cell identification was performed based on merged data from three sensors. However, as described in Section 3, rain cell identification is determined solely based on the PR data, by checking whether four or more pixels are connected. The infrared and microwave brightness temperature data are used only for deriving the physical parameters after the identification. Moreover, the MBR and BFE methods do not perform identification; they are fitting methods applied to already-identified rain cells.

Other instances with potentially misleading wording include:
- Lines 70–73 ("For the above...")
- Lines 134–135 ("To automatically...")
- Lines 301–304 ("By matching...")

Furthermore, many other parts of the manuscript incorrectly refer to MBR and BFE as identification methods. To avoid confusion, I recommend consistently referring to them as fitting methods.

Response: Thank you for your valuable suggestions! What you said is very correct. Just as you said, a rain cell is composed of connected rain pixels identified by PR, and then the MBR and BFE methods are used to fit these connected rain pixels to obtain the parameters of the rain cell. We simply abbreviate this process as the MBR and BFE methods for rain cell identification.

According to your suggestions, we have modified "identification method" in the full text to "fitting method" to avoid confusion in the expression. For example, the modified Lines 14-19 are as follows:

"In this study, based on the merged precipitation profile data, reflectivity and infrared data, and microwave brightness temperature data, …, rain cells were identified in the PR swath. For the identified valid rain cells, two fitting methods (the minimum bounding rectangle (MBR) and the best fit ellipse (BFE)) were applied to fit the external frame. Then, the geometric and physical parameters of rain cells were also calculated."

3.

In Section 5.2, while the two case studies help contextualize the dataset, the findings primarily reflect well-established knowledge and do not seem to introduce new insights. Please clarify what new synergies are expected by incorporating VIRS and TMI data into the dataset alongside the fitting of rain cells using the MBR and BFE methods. Additionally, since Section 5.2 feels somewhat redundant, I suggest minimizing the description of well-known results and moving them to the Introduction.

Response: Thank you very much for your reminder! The integration of rain cell geometric parameters with VIRS/TMI data yields the following synergistic effects:

1. The geometric parameters of rain cells can be obtained through the fitting methods, which can clearly characterize the morphological features of rain cells. Meanwhile, the visible and infrared channel signals of VIRS can provide cloud-top microphysical parameters, while the microwave brightness temperature of TMI can reflect intra-cloud hydrometeor information. The combination of such multi-source data enables us to deeply explore the interrelationships among geometric morphology parameters, precipitation intensity, and cloud microphysical parameters, thereby enhancing our understanding of precipitation systems.

2. The integration of multi-source data breaks the limitations of single observation methods in the past. By combining the observation data of active and passive instruments, we can build a richer database for precipitation detection. In particular, the relationship between the precipitation detected by PR and the brightness temperature of VIRS can be extended to the infrared channels of geostationary satellites. The corresponding research can not only optimize the precipitation

estimation of geostationary satellites, but also enrich precipitation inversion methods, thereby improving the ability to monitor precipitation.

3. The active observation of PR and the passive microwave observation of TMI can complement each other's advantages in precipitation detection. On the one hand, PR's high-precision precipitation results can serve as a reference for correcting the wide-swath TMI precipitation measurements on the same platform, optimizing the passive microwave inversion of precipitation results. On the other hand, the improved algorithm can be applied to microwave instruments of the same frequency band across different devices, achieving algorithm scalability and further enhancing the accuracy of precipitation inversion.

Additionally, we have revised the content in Section 5.2 by restructuring the wording for better clarity.

Minor comments:

Sections 3 and 4
They currently describe the algorithm separately; however, it would be more organized to combine them into a single section, such as a unified "Methods" section.

Response: Thank you very much for your suggestion. We have combined these two parts in the article.
3.1 The algorithm of rain cell identification
3.2 The definitions of rain cell parameters

Table 2
It should be clearly indicated which sensor each variable is derived from.

Response: Thank you! We have modified Table 2. The new table clearly indicates the sensor for each parameter.

Table 3
Many values listed in Table 3 are not referenced in the main text. It would be better to

remove unnecessary items. Regarding $\gamma_{max}$ and $\gamma_{avg}$, should these not differ between the MBR and BFE methods? Also, for parameters where the values are identical between MBR and BFE, combining them into a single column would improve readability.

Response: Thanks. The design of Table 3 has been restructured to align with the main text content and consolidate parameters with identical values into a single column. For the parameters $\gamma_{max}$ and $\gamma_{avg}$, as you pointed out, we recalculated and corrected the previous values. Since the results retain two significant figures, the displayed differences are not significant.

Lines 193–203

The content described in this paragraph might be easier to understand if explained directly with reference to Figure 1. If the authors wish to present specific numbers, it would be sufficient to mention them within the text. Therefore, I would suggest that Table 4 may not be necessary.

Response: Thank you for your nice suggestion. Table 4 presents the calculation results of physical parameters for two rain cell cases. Presenting them in the table can make the differences between the cases more intuitive. Based on your above suggestions, we have also redesigned Table 4 and added sensor labels for parameter detection, which corresponds to Table 2, thereby facilitating reader comprehension.

All the above have been modified in the revised manuscript. Thank you again.

---

## Author Response (AR3)

RC3:

The manuscript has greatly improved in terms of clarity and readability, and it now presents little risk of causing misunderstanding. Furthermore, with the release of two summer seasons' worth of data, the authors now meet the minimum requirement for conducting a long-term analysis. Based on these improvements, the reviewer recommends accept for publication after a very minor revision regarding terminology. No additional review is required at this stage.

minor comments:

1.
Line 196–209 There are some parenthetical usage that are uncommon in academic writing as follows.

"1.16 mm h-1 (A) and 2.31 mm h-1 (B), respectively": This phrasing is unfamiliar.

"CAF (convective area fraction to total precipitation area)": It is more conventional to write "convective area fraction to total precipitation area (CAF)" when introducing an abbreviation.

The purpose and style of parentheses differ between the sentences in L206 and L207. Consistency is recommended.

Response: Thank you very much for your suggestions! Regarding the issues such as parenthetical usage you raised, we have made revisions in the latest version of the manuscript. Thank you again for pointing out these detailed problems.

2.
Line251 Given the context, I recommend replacing "While" with "Because".

Response: Thank you for your suggestion! We have replaced "While" with "Because".

3.

Table 2 (VIRS entry), Section 4.2, L317, L333, etc.

In the context of visible or near-infrared channels such as VIRS, the values expressed on a scale from 0 to 1 are commonly referred to as "reflectance", not "reflectivity". The latter term is typically used in radar received echo power.

Response: Thank you for your revision suggestions! We have revised the relevant vocabulary throughout the entire text.

All the above have been modified in the revised manuscript. Thank you again.